# NO<sub>2</sub> pollution over India observed from space — the impact of rapid economic growth, and a recent decline

Andreas Hilboll<sup>1,2</sup>, Andreas Richter<sup>1</sup>, and John P. Burrows<sup>1</sup>

<sup>1</sup>Institute of Environmental Physics, University of Bremen, D-28359 Bremen, Germany <sup>2</sup>MARUM – Center for Marine Environmental Sciences, University of Bremen, D-28359 Bremen, Germany *Correspondence to:* Andreas Hilboll (hilboll@uni-bremen.de)

Abstract. The Indian economy has grown significantly during the past decades. Satellite-based remote sensing enables atmospheric pollution to be observed globally, in remote regions, and in regions where the infrastructure for air quality monitoring is limited. Here, we investigate the temporal evolution of tropospheric nitrogen dioxide ( $NO_2$ ) since the early 2000s, and correlate  $NO_2$  abundances with indicators of economic development, notably gross state domestic product and electricity generation expansity for all 35 Indian states and union territories.

5 capacity, for all 35 Indian states and union territories.

From 2003–2012, NO<sub>2</sub> pollution and economic growth are strongly correlated, leading to annual increases of up to 4.4%. This increase is strongest in states in East India having heavy industry. In 2012, the amount of tropospheric NO<sub>2</sub> reached a maximum; since then, tropospheric NO<sub>2</sub> pollution has stabilized or is even declining. While the Indian economy continues to grow, this decline in observed NO<sub>2</sub> values may be a result of a slow-down in Indian economic growth, combined with the

10 implementation of cleaner technology.

Additionally, we identify regional pollution sources such as individual steel smelters and the cement industry, which are severely degrading air quality. In Tamil Nadu, economic growth has not led to increasing  $NO_2$  columns, which we attribute to the investment in the development of renewable energy sources during the 2000s.

# 1 Introduction

Beginning in the 1950s, the Republic of India has seen tremendous growth of its population and economy. In the past 2–3 decades, this growth has been particularly dramatic. Between 2001 and 2011, the Indian population increased by 17.7%; the growth has been considerably more pronounced in urban (+31.8%) as compared to rural areas (+12.3%) (Office of the Registrar General & Census Commissioner, India, 2013). At the same time, the average population density in India increased from 325 to 382 km<sup>-2</sup>. Among the more populous states (population larger than 10,000,000), the growth rate varied between +4.9%
(Kerala) and +25.4% (Bihar); in virtually all states the urban population grew more rapidly than the rural population.

This strong increase in population has been accompanied by an enormous growth of the Indian economy; between the fiscal years (April–March) 2000/01 and 2013/14, the Indian gross domestic product (GDP) at factor cost and at constant prices has increased by ~144% (Reserve Bank of India, 2015).

5

Due to the high population density throughout India, and especially in the Indo-Gangetic Plain, the issue of poor air quality, resulting from anthropogenic emissions associated with this rapid economic growth, and its impact on human health has been recognized as an increasingly important societal issue. In this context, the nitrogen oxides ( $NO_x = NO + NO_2$ ) are important pollutants, because they are both toxic and participate in catalytic cycles producing the secondary pollutant and climate gas ozone ( $O_3$ ). Their oxidation in the gas phase by OH and reaction on wet surfaces or aerosols leads to nitric acid ( $HNO_3$ ) formation and acid deposition. More than half the  $NO_x$  emissions originate from anthropogenic sources, while those from the biosphere, lightning, and biomass burning play minor roles (Solomon et al., 2007). Consequently, maps of the tropospheric vertical column density ( $VCD_{trop}$ ) of  $NO_2$  are dominated by the regions of anthropogenic activity, involving fossil fuel combustion (see Fig. 1).

120 VCD<sub>trop</sub> NO<sub>2</sub> over India (SCIAMACHY, 2011/12) 36°N 100 30°N 80 24°N molec cm  $10^{14}$ 18°N 40 12°N 20 6°N 72°E 78°E 84°E 90°E 96°E - 0

**Figure 1.** Vertical NO<sub>2</sub> columns in the troposphere over India, as measured by the SCIAMACHY instrument, averaged over the 2011/12 fiscal year. Figure created with Cartopy (http://scitools.org.uk/cartopy) v0.14.2 using Natural Earth data.

Nitrogen dioxide  $(NO_2)$  has a strong and characteristically structured electronic vibrational rotational ultraviolet and visible spectrum, and its concentrations are such that its absorption is well observed in polluted region by remote sensing methods (Burrows et al., 2011). Since the mid-1990s, global data products of tropospheric NO<sub>2</sub> abundances retrieved from measurements by remote sensing instrumentation on satellite platforms are available. They are usually given as so-called vertical tropospheric columns (VCDs) which are defined as the vertical integral of the tropospheric NO<sub>2</sub> concentrations. These data

15 sets are retrieved from satellite-based nadir observations of the upwelling radiance in the blue part of the spectrum, using differential optical absorption spectroscopy (DOAS) (Platt and Stutz, 2008). The measurements are taken by the Global Ozone

Monitoring Experiment (GOME, on-board ERS-2) (Burrows et al., 1999), the SCanning Imaging Absorption spectroMeter for Atmospheric CHartographY (SCIAMACHY, on the ENVISAT platform) (Burrows et al., 1995; Bovensmann et al., 1999), the Ozone Monitoring Instrument (OMI, on-board NASA's Aura satellite) (Levelt et al., 2006), and the GOME-2 instruments (on-board Metop-A and Metop-B) (Callies et al., 2000). In this study, we use two data sets of VCD<sub>trop</sub> NO<sub>2</sub>, derived respectively from measurements of the SCIAMACHY instrument using the limb-nadir matching technique (Hilboll et al., 2013b), and a

data set combining five available satellite products (Hilboll et al., 2013a).

Tropospheric abundances of NO<sub>2</sub> over India have been the subject of numerous studies. For example, when GOME and SCIAMACHY measurements for the period 1996–2006 were analysed, trends over five larger regions of economic interest were derived. That study found (statistically insignificant) growth rates of between +1.0% and +2.5% annually (Ghude et al.,

- 2008). In addition, that study, by using a selection of 9 individual thermal power plants, obtained a linear dependence of the NO<sub>2</sub> column observed at the power plant locations on the installed power generation capacity. This trend analysis over regions of strong economic activity was later extended to the 1996–2012 period, when in the region of Delhi, the tropospheric NO<sub>2</sub> loading increased by  $4.05 \pm 0.84\%$  per year (Hilboll et al., 2013a). Surface NO<sub>x</sub> emissions were estimated from SCIAMACHY and OMI measurements of VCD<sub>trop</sub> NO<sub>2</sub>, yielding an average annual growth rate of 3.8% for the whole of India, in good
- agreement with emission inventories (Ghude et al., 2013a).

In the past three decades urbanization of the population has increased considerably and an ever-increasing part of the Indian population lives in large urban agglomerations. The three largest Indian megacities Delhi, Kolkata, and Mumbai have a combined population of almost 50 million inhabitants. The ambient concentrations of air pollutants have risen strongly in these major pollution centers. For example, tropospheric  $NO_2$  columns over Kolkata and Mumbai increased by 3.2 and 3.6 % yr<sup>-1</sup>,

- respectively, during 1996–2012, relative to the 1996 values (Hilboll et al., 2013a). A recent study focused on megacity emissions from traffic sources, because they account for 72% and 60% of total NO<sub>x</sub> emissions in Delhi and Mumbai, respectively. However, the authors found surprisingly low increases of ambient NO<sub>x</sub> pollution in the cities investigated, most notably in Kolkata (Gurjar et al., 2016).
- The strong economic growth of the Indian economy is reflected by an increasing demand for electricity, with the national 25 *five year plans* pushing for a continuous increase in the power generation capacity in order to facilitate the prescribed industrial 25 growth rate of 9% yr<sup>-1</sup>. More than half of the total Indian generation capacity comprises of coal fired power plants, resulting 26 in coal-based power generation being a major source of pollutant emissions with 80000–115000 annual premature deaths 27 attributed to this source of pollution (Guttikunda and Jawahar, 2014). Consequently, NO<sub>2</sub> columns over the states Chhattisgarh 28 and Odisha have increased by  $50 \pm 20\%$  from 2005 to 2015 (Krotkov et al., 2016).
- The seasonal cycle of tropospheric NO<sub>2</sub> columns over India is strongly influenced by the monsoon, with minimums occurring during pre-monsoon (Apr–May) and summer monsoon (Jun–Sep) seasons and a strong maximum during the dry season (Dec– Mar) (Ghude et al., 2008). It also strongly depends on the dominant NO<sub>x</sub> source in a given region; regions dominated by anthropogenic NO<sub>x</sub> emissions show a clear minimum in VCD<sub>trop</sub> NO<sub>2</sub> during August, followed by a steep increase up until December, and then a slight decrease, possibly leading to a local maximum in May or June, to the August minimum. Those
- 35 regions dominated by biomass burning or soil emissions also have their NO<sub>2</sub> minimum in summer (July/August), but the month

of maximum NO<sub>2</sub> columns varies (Ghude et al., 2013a). This strong winter maximum, however, has not been identified in a recent study using OMI measurements for the year 2007/08 (David and Nair, 2013). Finally, seasonal trend patterns similar to that for NO<sub>2</sub> have been observed for aerosol optical depth (AOD), with increases in AOD in the dry season, and inconsistent or weak trends during the monsoon (Babu et al., 2013).

5 This study is the first investigation of the long-term changes of the columnar tropospheric  $NO_2$  abundance over all 35 states and union territories of the Republic of India. We present the rates of change of  $VCD_{trop} NO_2$ , and compare and correlate these with and to socio-economic factors representative of the major anthropogenic emission sources: the gross state domestic product as an indicator of overall economic growth, the installed power generation capacity, directly linked to  $NO_x$  emissions caused by electricity generation, and the number of registered motor vehicles, indicative of traffic-related  $NO_x$  emissions.

#### 10 2 Methods

30

#### 2.1 Satellite measurements

Vertical tropospheric columns NO<sub>2</sub> were retrieved from measurements by the SCIAMACHY (ESA Envisat, 2002–2012), GOME-2 (MetOp-A 2006–present, MetOp-B 2012–present), and OMI (NASA Aura 2004–present) instruments. ENVISAT flew and MetOp-A and -B fly in sun-synchronous orbits in descending node, having a constant local equator crossing time of

15 10:00 LT and 09:30 LT, respectively (Burrows et al., 1995; Bovensmann et al., 1999; Callies et al., 2000). Aura is also in a sun-synchronous orbit but flies in ascending node with an equator crossing time of 13:30 LT.

The SCIAMACHY dataset used in this study is constructed using the limb-nadir matching technique (Hilboll et al., 2013b). Specifically, the analysis involves a) a spectral fitting with the DOAS procedure (Platt and Stutz, 2008; Richter and Burrows, 2002), b) subtraction of the stratospheric background NO<sub>2</sub> concentrations using limb-mode measurements by SCIAMACHY

itself (Rozanov et al., 2005), and c) radiative transfer simulations, using the model SCIATRAN (Rozanov et al., 2014), to account for the effective light path through the atmosphere.

Additionally, we use  $VCD_{trop} NO_2$  retrieved measurements from the GOME-2 instruments (Callies et al., 2000) on-board the European MetOp-A and MetOp-B satellites, and from the OMI instrument on-board NASA Aura (Levelt et al., 2006). The analysis for these two instruments is similar to the SCIAMACHY data analysis, the main differences being a different wave-

25 length window for the spectral fitting and the use of a chemistry-transport-model to account for stratospheric NO<sub>2</sub> background values (Richter et al., 2011; Hilboll et al., 2013a).

#### 2.2 State and union territory borders

This study works with the borders of Indian states and union territories as in place in 2014, using the Natural Earth dataset (Natural Earth, 2013). Therefore, the state *Telangana*, formed on 2 June 2014, is not separately included in the analysis but rather included in the *Andhra Pradesh* data.

4

# 2.3 Gross state domestic product

The *Reserve Bank of India* publishes the states' domestic products on an annual basis (Reserve Bank of India, 2016). For this study, we use Gross State Domestic Product at constant (2004/05) prices at factor cost, provided per fiscal year, i.e., for periods 1 April–31 March.

# 5 2.4 Population

For this study, we use gridded population density data for the year 2000 from version 1 of the Global Rural-Urban Mapping Project (GRUMP) (Center for International Earth Science Information Network, Columbia University et al., 2011).

# 2.5 Electricity production

The Indian Ministry of Power publishes monthly reports which contain information on the total installed electricity generation

capacity on a state-wise level, split up into sectors (state, private, central) and mode (coal, gas, diesel, nuclear, hydro, renewable). The reports are available at the ministry's website (Central Electricity Authority, Ministry of Power, 2016) for most years between 2003 and 2016.

When investigating the state-wise relative changes, some states / UTs (Daman & Diu, Dadra & Nagar Haveli, Sikkim) show implausible irregularities. While it seems obvious that the official datasets are incorrect in these cases, the affected states / UTs

only weakly contribute to Indian electricity production, and the results of this study are thus not affected by these irregularities.

# 2.6 Vehicle registrations

The Indian *Ministry of Road Transport and Highways* annually publishes state-wise numbers of registered motor vehicles in their so-called Road-Transport Yearbook, available at (Ministry of Road Transport and Highways, 2016). These yearbooks are available since the year 2006, and they also contain data for some selected large urban agglomerations.

Apparently, the data for Arunachal Pradesh show an implausible increase of the number of registered motor vehicles from 2011 to 2012.

# 2.7 Trend analysis

Monthly SCIAMACHY VCD<sub>trop</sub> NO<sub>2</sub> values have been analysed for their annual growth rates by fitting a linear function, plus a constant offset, plus a fourth-order harmonic seasonal cycle of constant amplitude and frequency (Hilboll et al., 2013a). All

25 absolute growth rates have been divided by the underlying value for the 2007/08 fiscal year to yield relative annual trends. A trend is not statistically significant if the 95% confidence interval of the slope parameter includes the 0.

5

#### 3 Results and Discussion

#### 3.1 Increase of tropospheric NO<sub>2</sub> pollution as observed from space

The tropospheric NO<sub>2</sub> load over the Indian subcontinent has strongly increased since the beginning of space-borne measurements in 1995 (Hilboll et al., 2013a). The VCD<sub>trop</sub> NO<sub>2</sub> retrieved from the hyper-spectral satellite sensors SCIAMACHY, GOME-2, and OMI consistently show a strong increase from 2003 onward, until they reach a maximum in 2012, with the amount stabilizing or declining thereafter (see Fig. 2, top left). When looking at the key polluted regions individually, this progression becomes even more distinct. For example, in the North Indian Plain, where the bulk of the Indian population resides (Fig. 2, top right), all sensors agree that the VCD<sub>trop</sub> NO<sub>2</sub> increased steadily from 2003 to 2011 and then appears to peak in 2012 before declining by ~15% between 2012 and 2015. In the coal mining and heavy industry laden states Chhattisgarh,

10 Jharkand, and Odisha (Fig. 2, bottom left), tropospheric NO<sub>2</sub> VCDs increased by ~30% between 2003 and 2012, with the amount stabilizing thereafter. In South India (Fig. 2, bottom right), the picture is not as clear. Here, the data from OMI show increasing tropospheric NO<sub>2</sub> VCDs from 2005 to 2012, with decreasing values thereafter; all other instruments show no clear trends.