# Peer review of "NO2 pollution over India observed from space — the impact of rapid economic growth, and a recent decline"

_Atmospheric Chemistry and Physics, 2017_

## Referee Comment (RC1) · Anonymous Referee #3 · 5 Apr 2017

Hilboll et al. presents an analysis of NO2 pollution changes over different regions of India and their socioeconomic drivers, by combining multiple satellite products and official socioeconomic data. I have a few suggestions as follows.

Multiple satellite products are used. Although some consistency is found in trend results (Fig. 1 and Fig. S1), there are clear quantitative differences among these products, especially after 2012. For example, the trend of OMI NO2 is clearly different from those in GOME2-a and GOME2-b (for all regions in Fig. 1 except North Indian Plain). For North Indian Plain, the OMI trend (Fig.1) is different from the DOMINO NO2 trend in Fig. S1. The large uncertainties in these satellite datasets make it difficult to conduct further linkage to socioeconomics. Is it possible to focus the analysis on regions

that multiple satellite products show quantitatively consistent trends? What are the uncertainties in trends from individual products considering retrieval and representative errors?

The NO2 growth rates are quantitatively significantly different from those in socioeconomic data, often by a factor of 10 (Table 1). It appears that chemistry, meteorology and/or other factors play major roles here. Can these factors be better accounted for in linking NO2 trends to emission trends? How are the roles of chemistry/meteorology in NO2 trends over India compared to the roles over other countries?

A statistical model is used to calculate the NO2 trends. Please discuss the model here briefly. Also, the model does not account for shift in seasonality when the pollution grew, which is important for fast changing pollution regions. Please discuss the caveat of this model.

Many regions are discussed here. A map defining all these regions will be extremely helpful for a general reader to understand the analyses.

The explanations in the last three paragraphs of Sect. 3.1 could be further improved. North Indian Plain also have large emissions from non-traffic sources such as power generation and industry. Can other factors be ruled out? The explanation for Chhattisgarh, Jharkand and Odisha is focused on power generation, how about other factors? Also, it is not clear why and how the monsoon signal is clear for South India but not for other regions. Can the changes over these regions be also found in OMI NO2 data? Overall, a region-specific analysis of major socioeconomic factors before discussing the causes of NO2 trends in these regions will much help the causation analysis.

The OMI NO2 data should be analyzed more intensively (e.g., in Figs 3,5,6 and Table 1), given its long temporal coverage (2004-present), different time of day (to help discuss the role of chemistry), and a higher resolution (to help reveal the hotspots). Comparing OMI with morning-time instruments will also help reveal the satellite uncertainty.

Fig.1 and S1 – starting the y-axis from a higher value (e.g., 15) than zero will help visualize the NO2 changes.

Fig. 3 – can you show results from OMI and quantitatively compare to GOME-2 results?

Fig. 6 – NO2 was flat (or even declined) from 2008 to 2011 while electricity and GSDP grew clearly. Why? How about the OMI NO2 data?

Table 1 – how about the NO2 trends from OMI?

Sect. 3.4 – where are the numbers (3e14 – 24e14 molec cm-2) from?

Conclusion – "This may imply that changes in meteorology or up to now not understood changes in tropospheric chemistry are also of significance." – given the uncertainty (especially after 2012), similar sentences addressing the roles of non-emission factors should be highlighted in the abstract.

---

## Referee Comment (RC2) · Anonymous Referee #1 · 7 Apr 2017

The study by Hilboll and co-authors focuses on changes in tropospheric NO2 pollution over India over the last 15 years. Using retrievals from 4 different satellite sensors, they present indications that NO2 pollution has been increasing between 2003 and 2013 over India, and some hints for a reduction 2014-2015. The authors relate the changes in the satellite NO2 columns directly to changes in presumed socio-economic drivers.

Major comments

1. There is insufficient detail on the satellite products used. It is not clear whether the SCIAMACHY, GOME-2, and OMI data used here have been validated. Neither it is clear whether the data from the SCIAMACHY and GOME-2 (MetOp-A and MetOP-B)

have been intercompared to check that they measure consistent columns over India. The paper should show for instance that GOME-2(A) and GOME-2(B) measure highly similar NO2 columns on the same day over India. Furthermore there is no information given on how the OMI NO2 product was generated, i.e. with a similar algorithm as GOME-2?

2. The direct attribution of trends to socio-economic drivers is questionable. There are many factors influencing the relationship between economic activities, subsequent emissions, and the measured NO2 columns. To name the most important ones: (a) sampling – measurements taken during the monsoon period (cloudy) are not suitable to detect the influence of emissions on NO2 (why not reject the monsoon period from the analysis?), (b) atmospheric chemistry – it is well known that the relationship between NOx emissions and the NO2 column depends on chemical and meteorological circumstances, and there may be differences between years that influence the relationship, especially when NOx emissions are changing, (c) errors in the socio-economic and in the satellite data – if one or both data sets suffer from time-dependent errors, it becomes difficult to argue that similar trends in both data sets allow direct attribution.

The authors seem to be aware of at least some of these issues, but do not address any of them other than making some remarks. I think they should make a much more convincing case for taking the satellite and socio-economic data at face value to make us believe there is a strong correlation between the two. In any case a more thorough analysis of sampling issues, intra-instrument consistencies and uncertainties is required, and the impact of variable meteorology and chemistry on the NO2 columns should be addressed with a model or otherwise.

3. The claim that the economy may grow without increased NO2 pollution on page 12 is very difficult to follow. The figure 6 shows very similar NO2 levels between 2003 and 2015 over Tamil Nadu, but also that energy production from fossil fuel combustion has increased strongly between 2011 and 2015. I can understand that if fossil electricity generation is driving NO2 pollution, we expect SCIAMACHY and GOME-2 to follow the

yellow line from 2003 to 2012. But, elsewhere in the paper, we are led to believe that NO2 increases when coal-burning starts, so why would this then not be the case over Tamil Nadu after 2012? It would help if the NO2 column values were given , and also the NOx emission contributions from the various sources.

Specific comments

P2, line 12: the Burrows et al. 2011-reference is not included in the reference list.

P2, L14: the vertical integration limits used in the retrieval should be given, i.e. what defines the tropopause?

Page 3, Lines 33-34: please explain why anthropogenic emissions are lowest in August.

Page 4, an indication on the accuracy and reliability of Indian socio-economic data would be welcome.

P5, section 2.7: there is no discussion on how uncertainty in the monthly mean is taken into account in the trend analysis. This should be done especially in view of the sometimes sparse sampling of SCIAMACHY data (between 0-5 measurements per month).

P6, Figure 2: it is not clear if the trends in the NO2 columns in Figure 2 have been obtained for retrievals without clouds. If so, do the bars represent proper 'annual means'? Or rather monsoon-filtered annual means?

P7, Figure 3: please include estimates of the uncertainties of the monthly means in the Figure.

P7, L12-13: Figure 3 a really strong seasonal cycle over India with a factor of 2-4 differences between winter and summer NO2 columns. It seems implausible that these differences can be explained from the difference in NOx-lifetime alone. Have the authors checked other reasons for this seasonal variability, e.g. emission variability or the

influence of air mass factors on the variability? Are slant column densities normalized with a geometric AMF also varying this strongly between Summer and Winter?

Page 8, Line 3-4: it is unclear why a "reduced growth rate" (of traffic-related NOx emissions) would contribute to NO2 decreases. If emissions are still growing, I'd only expect a decrease in NO2 concentrations if the emissions increase pushes the photochemical regime into the titration phase.

P8, L11-18: this paragraph on the delayed monsoon and its possible influence is merely speculating. My suggestion would be to analyse whether the decrease in 2014/2015 is due to the later monsoon in a more quantitative way via model simulations or other supporting data.

Page 8, line 12: pai?

Page 8, Line 24: it is unclear how the relative annual change rate in Figure 4 was calculated.

P8, L26-29: please indicate the cities of Ballari etc. on the large map of India. Not all readers will be familiar with the names of cities and regions in India.

P9, L2: with a sudden increase in 2010, how can you trust the linear regression trend analysis? This should be better explained.

P13, L5-7: this part is rather vague. Please clarify why this needs to be in the paper.

P13, section 3.4 seems like stating the obvious, and rather belongs in an introduction section.

---

## Author Comment (AC1) · 28 Sep 2017

**Reply to Reviewer Comment 1 (Anon. Ref. #3)**

Andreas Hilboll

September 27, 2017

We thank the anonymous reviewer #3 for the valuable suggestions how to improve our manuscript about $NO_2$ trends over India as observed from satellite and make its focus more clear to the reader.

> Hilboll et al. presents an analysis of NO2 pollution changes over different regions of India and their socioeconomic drivers, by combining multiple satellite products and official socioeconomic data. I have a few suggestions as follows. Multiple satellite products are used. Although some consistency is found in trend results (Fig. 1 and Fig. S1), there are clear quantitative differences among these products, especially after 2012. For example, the trend of OMI NO2 is clearly different from those in GOME2-a and GOME2-b (for all regions in Fig. 1 except North Indian Plain). For North Indian Plain, the OMI trend (Fig.1) is different from the DOMINO NO2 trend in Fig. S1. The large uncertainties in these satellite datasets make it difficult to conduct further linkage to socioeconomics. Is it possible to focus the analysis on regions that multiple satellite products show quantitatively consistent trends? What are the uncertainties in trends from individual products considering retrieval and representative errors?

We agree with the reviewer that the temporal variation is not always consistent between sensors and retrievals for all regions; this is indeed one of the major challenges in direct attribution of $NO_2$ changes to socio-economic factors. That being said, this manuscript's main goal is to report on these changes, most notably the surprising slow-down of observed $NO_2$ columns in spite of a growing economy and no sufficiently noteworthy changes in technology.

A robust, quantitative analysis of the potential *causal* relationship between

socio-economic factors and observed $NO_2$ columns is, while admittedly both interesting and important, however far beyond the scope of the present study. Therefore, we prefer to report on observed $NO_2$ and socio-economic changes for all Indian states, as the available satellite data have not been reported in such a manner before. This focus is reflected in the manuscript's title, which does not refer to socio-economic data at all.

Regarding the trend estimates, whenever we give quantitative trend estimates (as is the case in Tab. 1), trends have been marked as statistically significant if the 95% confidence interval for the slope parameter does not contain zero, as described in Sect. 2.7. The trend estimates are rather robust against retrieval errors, as trends from the different instruments agree well quantitatively. Regarding representative errors, we obviously rely on the assumption that over the course of the multi-year study period the spatial sampling of polluted and non-polluted areas inside one state averages out.

> The NO2 growth rates are quantitatively significantly different from those in socioeconomic data, often by a factor of 10 (Table 1). It appears that chemistry, meteorology and/or other factors play major roles here. Can these factors be better accounted for in linking NO2 trends to emission trends? How are the roles of chemistry/meteorology in NO2 trends over India compared to the roles over other countries? A statistical model is used to calculate the NO2 trends. Please discuss the model here briefly. Also, the model does not account for shift in seasonality when the pollution grew, which is important for fast changing pollution regions. Please discuss the caveat of this model.

The attribution of the observed $NO_2$ signal to individual socio-economic factors is indeed difficult. A quantitative analysis of this relationship is a whole study of its own and therefore outside the scope of the present manuscript, which aims to simply report on the temporal changes in $NO_2$ over the Indian states as observed from satellite.

The quantitative differences between change rates of $NO_2$ and socio-economic factors can come from a variety of sources. For example, the number of registered motor vehicles can only be a rough indicator for the incurring NOx emissions, as it does not take into account vehicle technology and fuel type. The same is true for the relationship between the generated electrical power and the incurring NOx emissions, which are not necessarily following a linear relationship, e.g., due to enhanced efficiency and advances in technology and flue gas cleaning. Similarly, GSDP in itself encompasses all sectors of the economy, including services which only have minor impact on NOx emissions.

All these reasons make a direct attribution and quantitative comparison of observed $NO_2$ columns to state-wide socio-economic factors difficult. Therefore, in this manuscript we have chosen to lay the focus on reporting on the temporal evolution of satellite-observed $NO_2$ over India, and to our knowledge, our study is the first to quantitatively describe its change at state level.

A detailed analysis of this relationship, maybe focused on one or two states, would be a very interesting follow-up to the present study. Lacking a good, up-to-date NOx emission inventory with reliable source attribution, such a study would involve the setup of a chemistry transport model for detailed sensitivity runs of a new, to-be-constructed emission model linking socio-economic activities to NOx emissions.

In the revised manuscript, we will try to give a better explanation of these relationships and make sure to make the focus of the present study clear to the reader, suggesting possible follow-on studies.

Regarding our statistical trend model: The model is presented in Sect. 2.7. and described in the referenced publication (Hilboll et al., 2013). Explicitly accounting for a changing amplitude of the seasonality component in the trend model usually leads to only marginally lower estimates of the linear trend component (see Sect 4.7.4 in Hilboll, 2014). In the revised manuscript, we will give more details about the trend model, including the caveat about changing seasonality.

> Many regions are discussed here. A map defining all these regions will be extremely helpful for a general reader to understand the analyses.

The revised manuscript will include a map detailing the locations discussed in the analysis.

> The explanations in the last three paragraphs of Sect. 3.1 could be further improved. North Indian Plain also have large emissions from non-traffic sources such as power generation and industry. Can other factors be ruled out? The explanation for Chhattisgarh, Jharkand and

> Odisha is focused on power generation, how about other factors?

According to the Ministry of Power's monthly reports, the source mix for electricity generation in the North Indian plain has not significantly changed in the last years. To our knowledge, there have been no changes in emission regulation and exhaust cleaning installation in power generation facilities in the North Indian Plain, which makes the electricity sector unlikely to be the cause of decreasing $NO_2$ concentrations. While GSDP increases in that area have lost some momentum compared to earlier years, they are still significant, giving no indication of declining emissions from industry sources.

In absence of effective regulation of coal-fired power generation emissions, and given the continuing commissioning of new power stations in Chhattisgarh, Jharkand and Odisha, it seems unlikely that the power sector would not be the driving force behind continuing $NO_2$ increases observed in that region. Of course, other factors cannot completely be ruled out, but lacking detailed and reliable NOx emission inventories at least at state level, it is impossible to be more confident.

In the revised manuscript, we will improve the description given in the paragraphs mentioned by the reviewer and give better account of our reasoning.

> Also, it is not clear why and how the monsoon signal is clear for South India but not for other regions. Can the changes over these regions be also found in OMI NO2 data? Overall, a region-specific analysis of major socioeconomic factors before discussing the causes of NO2 trends in these regions will much help the causation analysis. The OMI NO2 data should be analyzed more intensively (e.g., in Figs 3,5,6 and Table 1), given its long temporal coverage (2004-present), different time of day (to help discuss the role of chemistry), and a higher resolution (to help reveal the hotspots). Comparing OMI with morning-time instruments will also help reveal the satellite uncertainty.

For the revised manuscript, we will follow the reviewer's suggestion and include OMI $NO_2$ data in the analysis.

> Fig.1 and S1 – starting the y-axis from a higher value (e.g., 15) than zero will help visualize the NO2 changes.

We strongly believe that bar plots should always start at zero, in order not to mislead the viewer's reasoning. Therefore we will not change the y-axis in Figs. 1 and S1 (see, e.g., `https://flowingdata.com/2015/08/31/bar-chart-baselines-start-at-zero/` or `http://www.storytellingwithdata.com/blog/2012/09/bar-charts-must-have-zero-baseline`)

> Fig. 3 – can you show results from OMI and quantitatively compare to GOME-2 results?

The revised manuscript will contain an OMI version of Fig. 3, and the differences will be discussed.

> Fig. 6 – NO2 was flat (or even declined) from 2008 to 2011 while electricity and GSDP grew clearly. Why? How about the OMI NO2 data?

OMI and GOME-2 show very consistent results in the Tamil Nadu case, so that including OMI data in the plot does not give any new insight. However, for sake of completeness, we will include OMI data in Fig. 6 for the revised manuscript.

According to the Ministry of Power's monthly reports, the capacity of fossil fuel power generation in Tamil Nadu did not significantly increase in 2008-2011 (yellow line in Fig. 6). The total electricity generation was still strongly increasing, the bulk of the additional capacity coming from renewable sources. We describe in the manuscript how we believe that therefore, the $NO_2$ burden over Tamil Nadu could stay flat in spite of increasing GSDP and total electricity generation (renewable energy doesn't have NOx emissions).

> Table 1 – how about the NO2 trends from OMI?

In the revised manuscript, Table 1 will also include $NO_2$ trend results from OMI.

> Sect. 3.4 – where are the numbers (3e14 – 24e14 molec cm-2) from?

The numbers are from the Jena et al. (2015) article which is cited in the previous sentence. In the revised manuscript, we have rephrased the sentence to make this more clear.

> Conclusion – "This may imply that changes in meteorology or up to now not understood changes in tropospheric chemistry are also of significance." – given the uncertainty (especially after 2012), similar sentences addressing the roles of non-emission factors should be highlighted in the abstract.

We will add an according sentence to the abstract of the revised manuscript.

**References**

Hilboll, A., Richter, A., & Burrows, J. P. (2013). Long-term changes of tropospheric NO2 over megacities derived from multiple satellite instruments. Atmospheric Chemistry and Physics, 13(8), 4145–4169. `https://doi.org/10.5194/acp-13-4145-2013`

Hilboll, A. (2014). Tropospheric nitrogen dioxide from satellite measurements: SCIAMACHY limb/nadir matching and multi-instrument trend analysis (PhD thesis). Universität Bremen, Bremen. Retrieved from `http://nbn-resolving.de/urn:nbn:de:gbv:46-00103664-15`

Jena, C., Ghude, S. D., Beig, G., Chate, D. M., Kumar, R., Pfister, G. G., ... van der A, R. J. (2015). Inter-comparison of different NOX emission inventories and associated variation in simulated surface ozone in Indian region. Atmospheric Environment. `https://doi.org/10.1016/j.atmosenv.2015.06.057`

---

## Author Comment (AC2) · 28 Sep 2017

**Reply to Reviewer Comment 2 (Anon. Ref. #1)**

Andreas Hilboll

September 27, 2017

We thank the anonymous reviewer #1 for the valuable suggestions how to improve our manuscript about $NO_2$ trends over India as observed from satellite and make its focus more clear to the reader.

> There is insufficient detail on the satellite products used. It is not clear whether the SCIAMACHY, GOME-2, and OMI data used here have been validated. Neither it is clear whether the data from the SCIA-MACHY and GOME-2 (MetOp-A and MetOP-B) have been intercompared to check that they measure consistent columns over India. The paper should show for instance that GOME-2(A) and GOME-2(B) measure highly similar NO2 columns on the same day over India. Furthermore there is no information given on how the OMI NO2 product was generated, i.e. with a similar algorithm as GOME-2?

SCIAMACHY, OMI, and GOME-2/A have been shown to yield very consistent measurements over emission regions, e.g., by Hilboll et al. (2013).

$NO_2$ measurements from GOME-2 A and B have been extensively intercompared over several emission regions in Pinardi et al. (2015), showing excellent agreement between the sensors. While that study was performed using the operational EUMETSAT GOME-2 processor GDP4.8, the GDP retrieval algorithm has been modeled after the IUP-UB algorithm used in our study; consequently, the GOME-2 A/B retrievals for the IUP-UB retrieval algorithm can be expected to show the same level of agreement, which could be confirmed by internal tests (not published).

The OMI $NO_2$ product used in our study was generated with a DOAS retrieval based on a GOME-2 retrieval using a wide spectral window (425–497nm) presented in Richter et al. (2011), which was adapted for the special characteristics of the imaging instrument OMI (Richter et al., 2013).

The revised manuscript will contain a better description of the satellite datasets and references about all these points.

> The direct attribution of trends to socio-economic drivers is questionable. There are many factors influencing the relationship between economic activities, subsequent emissions, and the measured NO2 columns. To name the most important ones: (a) sampling – measurements taken during the monsoon period (cloudy) are not suitable to detect the influence of emissions on NO2 (why not reject the monsoon period from the analysis?), (b) atmospheric chemistry – it is well known that the relationship between NOx emissions and the NO2 column depends on chemical and meteorological circumstances, and there may be differences between years that influence the relationship, especially when NOx emissions are changing, (c) errors in the socio-economic and in the satellite data – if one or both data sets suffer from time-dependent errors, it becomes difficult to argue that similar trends in both data sets allow direct attribution. The authors seem to be aware of at least some of these issues, but do not address any of them other than making some remarks. I think they should make a much more convincing case for taking the satellite and socio-economic data at face value to make us believe there is a strong correlation between the two. In any case a more thorough analysis of sampling issues, intra-instrument consistencies and uncertainties is required, and the impact of variable meteorology and chemistry on the NO2 columns should be addressed with a model or otherwise.

We are well aware of the complexity of the relationship between economic activities, emissions, and measured $NO_2$ columns and agree with the reviewer's list of "most important" influencing factors.

That being said, this manuscript's main goal is to report on the changes in satellite-observed $NO_2$ over India, most notably the surprising slow-down of observed $NO_2$ columns in spite of a growing economy and no sufficiently noteworthy changes in technology. This focus is reflected in the manuscript's title, which does not refer to socio-economic data at all. As part of this report, we find it important to point out the strong *correlation* between this $NO_2$ increase and various socio-economic factors, which does not necessarily imply *causation*. A robust, quantitative analysis of the potential *causal* relationship between socio-economic factors and observed $NO_2$ columns is, while admittedly both interesting and important, however far beyond the scope of the present study.

In the revised manuscript, we will make the focus of the study more clear, give care to avoid implying clear causal, quantitative relationships between economy and $NO_2$, and provide a better description of the potential caveats of direct attribution.

> The claim that the economy may grow without increased NO2 pollution on page 12 is very difficult to follow. The figure 6 shows very similar NO2 levels between 2003 and 2015 over Tamil Nadu, but also that energy production from fossil fuel combustion has increased strongly between 2011 and 2015. I can understand that if fossil electricity generation is driving NO2 pollution, we expect SCIAMACHY and GOME-2 to follow the yellow line from 2003 to 2012. But, elsewhere in the paper, we are led to believe that NO2 increases when coal-burning starts, so why would this then not be the case over Tamil Nadu after 2012? It would help if the NO2 column values were given, and also the NOx emission contributions from the various sources.

We agree with the reviewer that in view of the strong increase on coal-fired electricity production in recent, the lack of $NO_2$ increases after 2011 is surprising. However, $NO_2$ pollution by coal-fired power plants is mostly a local phenomenon of which, given the location of the newly constructed power plants close to the coast or state border, only part (i.e., the part being over land and inside state territory) can influence the $NO_2$ levels which our study attributes to the state of Tamil Nadu. As we lay out in the manuscript, we believe that these changes are not high enough (yet) to significantly influence the state-wide averages. In the revised manuscript, we will make this reasoning more clear to the reader and formulate this fact as the open research question that it is. Also, Fig. 6 will be given a second y-axis giving the $NO_2$ columns from the GOME-2 instrument.

> P2, line 12: the Burrows et al. 2011-reference is not included in the reference list.

The updated manuscript will contain Burrows et al. (2011) in the reference list.

> P2, L14: the vertical integration limits used in the retrieval should be given, i.e. what defines the tropopause?

The DOAS method measures the *vertically integrated* $NO_2$ amount in the atmosphere, which is only subsequently corrected for the influence of stratospheric $NO_2$ (which comes from independent measurements or from an external data source, i.e., model data). The definition of the tropopause becomes important only in this post-processing step. It should be noted that the actual tropopause height is not critical in $NO_2$ retrievals because of the very low $NO_2$ concentrations in the altitude region between 8km and 20km.

That being said, methods to define the tropopause vary between data products, from a simple constant or latitude-dependent assumption to accurate calculations from meteorological model data. In our case, we use ECMWF ERA-Interim reanalysis data to calculate the tropopause from the potential vorticity fields; details are given in the referenced Hilboll et al. (2013b) publication.

Since the mentioned sentence (p. 2, l. 14) belongs to the introduction of our manuscript and does not apply to any actual data set in particular, we believe that the integration limits should not be mentioned at that place.

The revised manuscript will however mention our method to define the tropopause in the *Methods* section.

> Page 3, Lines 33-34: please explain why anthropogenic emissions are lowest in August.

In the manuscript, we state that the *measured tropospheric column densities $VCD_{trop}$* are lowest in August. The seasonality of $NO_2$ columns over India is mostly driven by meteorology, i.e., the minimum in August is mostly caused by the monsoon (see ul Haq et al., 2015, and Ghude et al. 2013) and by the dependence of $NO_2$ life-time on photochemistry (again, leading to lower $NO_2$ concentrations in summer, when the sun is high). This is especially pronounced in regions of strong (anthropogenic) emissions, as only there a significant amount of $NO_2$ is released to the atmosphere which can actually be washed out.

> Page 4, an indication on the accuracy and reliability of Indian socioeconomic data would be welcome.

All Indian socio-economic indicators used in this study have been collected from official government sources. The revised manuscript will contain a note about their reliability.

P5, section 2.7: there is no discussion on how uncertainty in the monthly mean is taken into account in the trend analysis. This should be done especially in view of the sometimes sparse sampling of SCIA-MACHY data (between 0-5 measurements per month).

The uncertainty in the monthly mean is not taken into account in the trend analysis. However, the sparse sampling of SCIAMACHY data can be assumed to not pose any limitation on our trend estimates, as measurements by GOME-2 and OMI, which have different, independent sampling patterns, yield similar results.

Regarding sampling due to cloud cover, Wonsick et al. (2009) could show that the peak amplitude of the diurnal cycle of cloud cover over India is rather low (below 30%), and that between the instruments' measurement times there is no large variation. While this does not rule out the possibility of our trend estimates being influenced by sampling issues caused by the cloud cover, it seems unlikely that this issue would cause any systematic effect on our results.

In the revised manuscript, we will give a short account of the sampling issue.

Also, the revised manuscript will include trend estimates for $NO_2$ columns retrieved from the OMI instrument as an additional dataset, showing the robustness of the trend estimate results.

P6, Figure 2: it is not clear if the trends in the NO2 columns in Figure 2 have been obtained for retrievals without clouds. If so, do the bars represent proper 'annual means'? Or rather monsoon-filtered annual means?

Figure 2 shows annual mean $NO_2$ columns, which are calculated as average of the 12 monthly means of cloud-filtered $NO_2$ $VCD_{trop}$ over the respective regions. This means that each month contributes equally to the displayed annual mean, i.e., monsoon months are not filtered out.

P7, Figure 3: please include estimates of the uncertainties of the monthly means in the Figure.

The uncertainties of the monthly means are hard to quantify, as very different factors (spatial sampling, temporal sampling, sampling for meteorological condition due to cloud filter, intra-monthly variability of $NO_2$ VCDs, . . . ) contribute and a thorough analysis of their individual importance and interdependence can only be estimated with complex model sensitivity runs. Given the highly uncertain uncertainty estimates, we therefore choose to not give any quantitative estimates.

While the quantification of the uncertainties is a very interesting (and we agree, also important) study, it lies clearly outside the scope of the present article, which wants to report on $NO_2$ increases over India as observed from space.

That being said, the trend method has proven to be robust against outliers caused by measurement noise (Hilboll 2014).

P7, L12-13: Figure 3 a really strong seasonal cycle over India with a factor of 2-4 differences between winter and summer NO2 columns. It seems implausible that these differences can be explained from the difference in NOx-lifetime alone. Have the authors checked other reasons for this seasonal variability, e.g. emission variability or the influence of air mass factors on the variability? Are slant column densities normalized with a geometric AMF also varying this strongly between Summer and Winter?

$NO_2$ columns over India have been reported to show a strong seasonal cycle in previous studies (see, e.g., Ghude et al. 2013). The especially pronounced seasonal cycle is a known feature of the IUP-UB $NO_2$ product, which is partly caused by the used AMFs which are derived from a monthly climatology of $NO_2$ vertical profiles derived from the MOZART-2 model. However, it has been shown (see Hilboll 2014) that the seasonal cycle is only being enhanced by these AMFs, as SCDs normalized with a geometric AMF also show a pronounced seasonal cycle.

That being said, one should note that the strong seasonal cycle does not impact significantly on the estimated annual change rates. The amplitude of the seasonality is one of the fit parameters in our trend model and has

been shown (Hilboll 2014) to not significantly impact on the resulting $NO_2$ trends.

> Page 8, Line 3-4: it is unclear why a "reduced growth rate" (of traffic-related NOx emissions) would contribute to NO2 decreases. If emissions are still growing, I'd only expect a decrease in NO2 concentrations if the emissions increase pushes the photochemical regime into the titration phase.

We agree with the reviewer's remark, and will re-phrase this paragraph in the revised manuscript.

> P8, L11-18: this paragraph on the delayed monsoon and its possible influence is merely speculating. My suggestion would be to analyse whether the decrease in 2014/2015 is due to the later monsoon in a more quantitative way via model simulations or other supporting data.

We thank the reviewer for the honest criticism of our, admittedly speculative, argument. While we believe that this is a very interesting aspect, performing dedicated model simulations for the investigation of this point is however outside the scope of the present article, which wants to mainly report on the increase of $NO_2$ over India as observed from satellite. In the revised manuscript, we will therefore give less emphasis on this and suggest future studies be performed to investigate this aspect.

> Page 8, line 12: pai?

This refers to the Monsoon reports for 2014 and 2015 by Pai and Bhan; we will fix the citation in the revised manuscript.

> Page 8, Line 24: it is unclear how the relative annual change rate in Figure 4 was calculated.

The trend analysis is already briefly discussed in Section 2.7. In the revised manuscript, we will add a reference to that section to the Figure caption.

> P8, L26-29: please indicate the cities of Ballari etc. on the large map of India. Not all readers will be familiar with the names of cities and

> regions in India.

In the revised manuscript, the locations mentioned in the text will be indicated in a map, where feasible.

> P9, L2: with a sudden increase in 2010, how can you trust the linear regression trend analysis? This should be better explained.

A linear regression trend analysis can only give an average growth rate of the study period. In case of newly constructed emission sources, e.g., steel furnaces or power plants, the resulting slope of the regression line depends just as much on the length of the study period as on the actual increase in $NO_2$ concentrations.

That being said, since our study uses the same time period for all linear regression trends, the results do allow comparing *average* $NO_2$ increases between different locations.

> P13, L5-7: this part is rather vague. Please clarify why this needs to be in the paper.

We have removed this passage from the revised manuscript.

> P13, section 3.4 seems like stating the obvious, and rather belongs in an introduction section.

We agree; the revised manuscript will contain the contents of Section 3.4 in the introduction.

**References**

Burrows, J. P., Platt, U., & Borrell, P. (2011). The Remote Sensing of Tropospheric Composition from Space (1st ed.). Heidelberg: Springer.

Ghude, S. D., Kulkarni, S. H., Jena, C., Pfister, G. G., Beig, G., Fadnavis, S., & van der A, R. J. (2013). Application of satellite observations for identifying regions of dominant sources of nitrogen oxides over the Indian Subcontinent. Journal of Geophysical Research: Atmospheres, 118(2), 1075–1089.

https://doi.org/10.1029/2012JD017811

Hilboll, A., Richter, A., & Burrows, J. P. (2013). Long-term changes of tropospheric NO2 over megacities derived from multiple satellite instruments. Atmospheric Chemistry and Physics, 13(8), 4145–4169. https://doi.org/10.5194/acp-13-4145-2013

Hilboll, A., Richter, A., Rozanov, A., Hodnebrog, Ø., Heckel, A., Solberg, S., ... Burrows, J. P. (2013b). Improvements to the retrieval of tropospheric NO2 from satellite – stratospheric correction using SCIAMACHY limb/nadir matching and comparison to Oslo CTM2 simulations. Atmospheric Measurement Techniques, 6, 565–584. https://doi.org/10.5194/amt-6-565-2013

Hilboll, A. (2014). Tropospheric nitrogen dioxide from satellite measurements: SCIAMACHY limb/nadir matching and multi-instrument trend analysis (PhD thesis). Universität Bremen, Bremen. Retrieved from http://nbn-resolving.de/urn:nbn:de:gbv:46-00103664-15

Pinardi, G., Lambert, J.-C., Yu, H., De Smedt, I., Granville, J., Van Roozendael, M., & Valks, P. (2015). O3M SAF Validation Report (O3M SAF Validation Report No. SAF/O3M/IASB/VR/NO2). Retrieved from http://acsaf.org/docs/vr/Validation_Report_NTO_OTO_DR_NO2_GDP48_Nov_2015.pdf

Richter, A., Begoin, M., Hilboll, A., & Burrows, J. P. (2011). An improved NO2 retrieval for the GOME-2 satellite instrument. Atmospheric Measurement Techniques, 4, 1147–1159. https://doi.org/10.5194/amt-4-1147-2011

ul-Haq, Z., Tariq, S., & Ali, M. (2015). Tropospheric NO2 trends over South Asia during the last decade (2004-2014) using OMI data. Advances in Meteorology. Retrieved from http://www.hindawi.com/journals/amete/aip/959284/

Wonsick, M. M., Pinker, R. T., & Govaerts, Y. (2009). Cloud Variability over the Indian Monsoon Region as Observed from Satellites. Journal of Applied Meteorology and Climatology, 48(9), 1803–1821. https://doi.org/10.1175/2009JAMC2027.1